# Ovarian Drilling: Back to the Future

**DOI:** 10.3390/medicina58081002

**Published:** 2022-07-27

**Authors:** Antonio Mercorio, Luigi Della Corte, Maria Chiara De Angelis, Cira Buonfantino, Carlo Ronsini, Giuseppe Bifulco, Pierluigi Giampaolino

**Affiliations:** 1Department of Neuroscience, Reproductive Sciences and Dentistry, School of Medicine, University of Naples Federico II, 80138 Naples, Italy; antoniomercorio@gmail.com (A.M.); cirabuonfantino@gmail.com (C.B.); giuseppe.bifulco@unina.it (G.B.); 2Department of Public Health, School of Medicine, University of Naples Federico II, 80138 Naples, Italy; m.chiaradeangelis@gmail.com (M.C.D.A.); pgiampaolino@gmail.com (P.G.); 3Department of Woman, Child and General and Specialized Surgery, School of Medicine and Surgery, University of Campania Luigi Vanvitelli, 81100 Napoli, Italy; carlo.ronsini90@gmail.com

**Keywords:** polycystic ovary syndrome, infertility, ovarian drilling, laparoscopy

## Abstract

Polycystic ovary syndrome (PCOS) is the leading cause of anovulatory infertility. The complex metabolic dysregulation at the base of this syndrome often renders infertility management challenging. Many pharmacological strategies have been applied for the induction of ovulation with a non-negligible rate of severe complications such as ovarian hyperstimulation syndrome and multiple pregnancies. Ovarian drilling (OD) is currently being adopted as a second-line treatment, to be performed in case of medical therapy. Laparoscopic ovarian drilling (LOD), the contemporary version of ovarian wedge resection, is considered effective for gonadotropins in terms of live birth rates, but without the risks of iatrogenic complications in gonadotropin therapy. Its endocrinal effects are longer lasting and, after the accomplishment of this procedure, ovarian responsiveness to successive ovulation induction agents is enhanced. Traditional LOD, however, is burdened by the potential risks of iatrogenic adhesions and decreased ovarian reserve and, therefore, should only be considered in selected cases. To overcome these limits, novel tailored and mini-invasive approaches, which are still waiting for wide acceptance, have been introduced, although their role is still not well-clarified and none of them have provided enough evidence in terms of efficacy and safety.

## 1. Introduction

Polycystic ovary syndrome (PCOS) is a complex endocrinopathy, characterized by oligoanovulation, hyperandrogenism, and an abnormal ovarian morphology characterized by multiple small subcapsular follicular ‘cysts’ [1,2].

PCOS is the leading cause of anovulatory infertility, accounting for nearly 80% of all cases [3]. Many therapeutic strategies have been applied for inducing ovulation in these patients: clomiphene citrate (CC) is considered a convenient and economic choice; however, 15% to 40% of patients are CC-resistant and deserve to be treated with gonadotropins or other medical ovulation-induction agents. These medications are not always successful, can be time-consuming, and can cause adverse events such as multiple pregnancies or require cycle cancellation due to an excessive response. Ovarian drilling (OD) is a second-line treatment to be considered in case of medical therapy failure. OD results in an overall spontaneous ovulation rate of 30–90% and final pregnancy rates of 13–88% [4]. Over the years, different OD techniques have been described in the literature. Most of them are intended to overcome some negative aspects of the traditional laparoscopic approach such as the risk of adhesion development, ovarian reserve damage, and abdominal wall trauma while maintaining comparable reproductive outcomes.

The endeavor of this narrative review is to provide an update on the available evidence on the surgical management of PCOS-related anovulatory infertility. After a short focus on the main indications and limits of medical therapy, the current body of literature regarding the role of traditional laparoscopic ovarian drilling (LOD), entailing its different preoperative, operative, and postoperative aspects, will be summarized. Finally, this paper will approach the new available therapeutic modalities of OD and their current limitations.

## 2. Materials and Methods

An electronic database search (PubMed, Medline and Embase) was performed up to March 2022. A search algorithm was developed incorporating the terms “polycystic ovary syndrome”, “infertility”, “clomiphene citrate”, “gonadotropin”, “laparoscopy”, “ovarian drilling”, “transvaginal hydrolaparoscopy”, and “ovulation induction”. The search analysis was carried out by two coauthors (A.M. and L.D.C.).

Original studies evaluating the different surgical strategies available for the treatment of PCOS-associated anovulatory infertility, their association with reproductive outcomes and current limitations were critically reviewed. Only articles in the English language were included. The reference lists were systematically reviewed to identify other studies for potential inclusion in this narrative review.

## 3. Medical Therapy: A Brief Summary

Anovulation treatment is a major challenge for clinicians: many therapeutic strategies have been applied over the years but, to date, the ideal one is yet to be identified. A “stepwise approach” for the management of PCOS-related infertility is highly recommended. Lifestyle adjustment is the first intervention to be implemented: women with PCOS are frequently overweight; therefore, weight loss is recommended as primary therapy for anovulatory PCOS women with a body mass index (BMI) ≥ 30 kg/m^2^. Bodyweight loss may promote spontaneous ovulation by decreasing hyperandrogenism, LH levels, and insulin resistance [5,6].

If unsuccessful, clomiphene citrate (CC) therapy has a major role in the induction of ovulation for patients seeking pregnancy. The dosage of CC can be increased by 50 mg for the subsequent cycle in case of persistent anovulation. Using this approach, 50% of patients will ovulate in response to 50 mg, almost 20% will ovulate with 100 mg, and 10% respond to 150 mg [7]. Women demonstrating an absence of ovulation after clomiphene administration at doses up to 150 mg/day should be considered “clomiphene resistant” [8]. These patients are more likely to be overweight, insulin-resistant, and hyperandrogenic than those who do respond [9,10]. Among women with successful ovulatory treatment for 6–9 cycles, the pregnancy rate is disappointingly low (30 to 50%). The discrepancy noted between good ovulation rates and lower pregnancy rates is probably due to the anti-estrogenic effects of CC on the endometrium, cervical mucus, and the hypersecretion of LH leading to premature luteinization [7]; the term “clomiphene-failure” is used to indicate the inability to conceive after CC treatment. In cases of CC resistance/failure, the addition of metformin, tamoxifen, and rosiglitazone can significantly increase ovulation and pregnancy rate [11,12].

As an alternative to CC, the aromatase inhibitor letrozole has been recommended as the first-line pharmacological treatment for ovulation induction in women with PCOS anovulatory infertility [13]. However, due to its prohibitive cost, its off-label use in many countries and its controversial adverse effects including gastrointestinal disturbances, hot flushes, headache, and back pain, CC is still the most commonly used oral ovulatory drug [14].

In this regard, not to be overlooked is the safety alarm raised by several authors about an increased incidence of birth defects in patients treated with CC or letrozole [15,16,17].

New PCOS guidelines published in 2018 recommend injectable gonadotropins used alone or with the addition of metformin as second-line pharmacological agents to induce ovulation in women following unsuccessful treatment with first-line oral ovulation induction agents. In a multi-center randomized clinical trial (RCT) comparing CC versus gonadotropins in anovulatory women with PCOS who were therapy-naïve, significantly higher clinical pregnancy rates in the gonadotropins group were reported [13]. Despite its efficacy, the necessity of daily injections, the risk of adverse events such as multiple pregnancies or cycle cancellation due to an excessive ovarian response (ovarian hyperstimulation syndrome—OHSS), and the need for intensive ultrasound monitoring make gonadotrophins treatment an expensive, inconvenient, and time-consuming choice [18]. The replacement of the standard protocol associated with an unacceptable rate of multiple pregnancies and increased risk of OHSS with the low-dose protocol has significantly reduced but not abolished these unwanted effects [19,20,21].

## 4. Laparoscopic Ovarian Drilling

In 1935, Stein and Leventhal reported the first successful treatment of infertile PCOS women through laparotomic “wedge resection” procedure [22]. Regardless of these promising results, the surgical approach was outdone by pharmacological treatment due to the high risk of pelvic adhesions following surgery. The surgical treatment of CC-resistant PCOS cases improved remarkably with the introduction of a minimally invasive approach: laparoscopic ovarian drilling (LOD) [23]. The exact mechanism by which small perforations using heat or a laser result in follicular growth and ovulation is yet to be elucidated and it is not known whether a prevalent action is exerted through a direct effect on the ovary or through a systemic endocrine mechanism. The most plausible mechanism is that the thermal destruction of ovarian follicles and a part of the ovarian androgen-producing stroma results in the reduction in local and serum androgens, re-establishing an intrafollicular environment more convenient for normal follicular maturation and ovulation and a secondary rise in follicle-stimulating hormone (FSH) levels. In addition, the release of a cascade of local growth factors such as insulin-like growth factors interacting with FSH, following a surgery-mediated increase in ovarian blood in response to thermal injury, has been suggested to allow follicular growth and subsequent ovulation [24]. Further possible mechanisms are the decrease in anti-Müllerian hormone (AMH) concentrations and the production of “holes” in the very thick cortical wall of the polycystic ovary [25]. The efficacy of ovarian drilling is widely variable in the literature: in a comprehensive review, ovarian drilling is deemed to restore fertility in 20–64% of women with PCOS previously suffering from anovulatory infertility who did not respond to CC treatment; 70% of pregnancies occurred in the first 6 postoperative months [26]. The surgical approach has some advantages in comparison to medical treatment [27,28,29,30]: no significant differences were found with respect to live birth rate and miscarriage between LOD and gonadotropins or other medical treatment in women resistant to CC, with the advantage of spontaneous mono-ovulation without the need for intensive monitoring in order to minimize the risks of multiple pregnancies or OHSS. For a similar success rate, LOD might be indicated as a second-line therapy instead of gonadotrophins to avoid gonadotropin-related adverse events (Table 1).

Although no differences in pregnancy rate were reported even when LOD was compared to CC treatment as first-line therapy [31], there is insufficient evidence to support the use of LOD as first-line therapy except if laparoscopy is indicated for another reason (e.g., a diagnostic evaluation for tubal patency). Beyond the reduced risk of OHSS and multiple pregnancies, further advantages of LOD over medical treatment should be considered:(a)Even if the effect of ovarian diathermy is not permanent and should, therefore, be reserved for infertile women, the positive reproductive outcome seems to last for several years in many women with the advantage of repeated spontaneous ovulation and further pregnancies as opposed to multiple rounds of ovulation induction [32].(b)The increased responsiveness of the ovary to CC or gonadotropin medical therapy after LOD failure can be of invaluable help before proceeding to assisted reproductive therapy [33], mainly in vitro fertilization (IVF), which is considered an effective treatment option in anovulatory PCOS patients who do not become pregnant with ovarian drilling [34]. The stimulation in PCOS women is typically more difficult than healthy women and often experience a higher cycle cancellation rate when compared with normo-responders. The incidence of OHSS has been reported to be statistically significantly lower among patients with antecedent surgical treatment [35].(c)LOD is considerably less expensive than ovulation induction with gonadotropins: a single treatment results in several mono-ovulatory cycles, whereas one course of gonadotropin therapy yields a single ovulatory cycle with an inherent cost for intensive monitoring. The higher incidence of multiple pregnancies incurs extra costs in those who conceive with gonadotrophins [36,37].

Finally, the aforementioned lower cycle cancellation rates in patients later submitted to IVF as well as the reduced incidence of OHSS contribute to lessening indirect costs.

## 5. Intra- and Post-Operative Potential Risk of Ovarian Drilling

Among the possible adverse effects, surgical morbidity of this procedure should not be underestimated since it is frequently performed in overweight or obese women. In addition, specific concerns exist regarding the eventuality of iatrogenic adnexal adhesions and the reduction in ovarian reserve. The rate of adnexal adhesions differs widely in various studies ranging from 19 to 60%, mostly consisting of mild to moderate severity that seems to not affect pregnancy [38,39]. However, due to this potential risk, LOD should be performed by fully trained laparoscopic surgeons, thereby reducing the likelihood of thermal damage, risk of adhesion, or injury to the neighboring viscera. Numerous strategies should be adopted (Table 2):

(a)Ovaries should be lifted before energy application and should be cooled by normal saline immediately after the procedure;(b)Peritoneal lavage and artificial ascites [40] or application of adhesion barriers should be considered even if certain efficacy in preventing adhesions and improving reproductive outcomes has not been proven [41];(c)The number of drills and amount of diathermy should be reduced to the lowest effective dose.

Different types of energy sources and methods are reported in the literature for the accomplishment of the procedure: monopolar diathermy, using an insulated unipolar needle electrode with a non-insulated distal end measuring 1–2 cm, is the most widely used technique, although few authors have reported similar ovulation and pregnancy rates with a monopolar hook electrode or bipolar energy [42]. The use of the harmonic scalpel was found in a randomized study to be just as effective as the Nd-YAG laser, although the latter has been found to be more prone to preventing adhesion owing to lower thermal penetration by the cone-shaped lesions of laser drilling [43]. Lack of consensus exists regarding the amount of electrosurgical energy and the optimal number of punctures holes to achieve maximum efficacy: reducing the thermal energy (<300 J/ovary) reduces the chances of ovulation and pregnancy, while higher thermal doses (>1000 J/ovary) may result in extensive tissue destruction without additional improvement in outcomes [44]. Exemplary is the case of ovarian atrophy following high-energy drilling (eight coagulation points at 400 W for 5 s) [45].

Regarding the puncture holes, so far, it has been established that the administration of only two punctures in each ovary is insufficient to induce ovulation and more than eight punctures increase the occurrence of postoperative pelvic adhesions and ovarian reserve damage; it is inconclusive whether there is a difference in relation to clinical and reproductive outcome following greater than four ovarian punctures in each ovary [46]. In initial studies, it was hypothesized that the greater the energy, the higher the success rate. Subsequently, lower temperatures with a fixed number of drilled points have been reported to be successful with minimal ovary injury effect. To date, it can be assumed that the ovulatory and pregnancy rate is dose-dependent up to a maximum thermal energy of 640 J/ovary, which can, therefore, be considered the lowest amount for effective diathermy [47]. Most surgeons adopt this method by performing four punctures bilaterally for a depth of 3–4 mm diameter with a mixed current in the monopolar electrosurgical needle, each for 4 s at 40 W (rule of 4), delivering 640 J of energy per ovary; however, such fixed dosage may be inappropriate in women with enlarged ovaries and, in this regard, a new methodology of tailoring the energy according to the ovarian volume has been proposed.

Reduced ovarian reserve is a great potential complication of this procedure. PCOS is characterized by serum AMH 2–4-fold in excess compared with healthy women. AMH concentration correlates with the number of small antral follicles, and ovaries in PCOS patients exhibit an increased number of preantral and small antral follicles. LOD determines a marked decline in these abnormally elevated AMH concentrations. Whether this drop can be explained as a temporary normalization of ovarian markers with subsequent recovery rather than a permanent reduction in ovarian reserve remains to be elucidated [48]. At present, due to the heterogeneity of the studies, there is insufficient evidence to determine the impact of LOD on ovarian reserve. The optimal balance between benefits and potential adverse effects of ovarian cauterization such as surgical morbidity, iatrogenic adhesion and ovarian reserve damage must be achieved with proper patient selection. Several prognostic factors predict successful outcomes and should be considered before choosing this surgical option [49]. Factors increasing the efficacy of this technique are a normal body mass index (BMI), high LH concentration (>10 UI/L), short infertility duration, and age less than 35 [50]. Patients with BMI values greater than 35 kg/m^2^ obtain lower ovulation rates (13%) compared to patients with BMI between 29 and 34 kg/m^2^ (46%) and those with BMI < 29 kg/m^2^ (57%) [51]. Possible predictors of poor outcomes are hyperinsulinemia, elevated AMH levels and high testosterone serum levels [52]. Indeed, among women with high serum testosterone levels (≥4.5 nmol/L), the ovulation rate was 10%, significantly lower than among those with moderate levels of testosterone (2.6 to 4.4 nmol/L), which was 48%, and normal testosterone (2.6 nmol/L), with a 56% ovulation rate [53].

## 6. Alternative Options to Traditional Surgical Ovarian Drilling

Unilateral ovarian drilling (ULOD) has been proposed as a modification of the standard LOD methodology with encouraging results [42]. ULOD induces activity in both ovaries and minimizes procedure time; the lack of significant differences in terms of clinical and biochemical response, ovulation rate, and pregnancy rate if compared with conventional bilateral LOD (BLOD) let us consider this technique suitable for CC-resistant PCOS [54,55,56]. At the moment, due to the paucity of available studies, it is uncertain if the equivalent reproductive outcome is associated with a lower risk of post-operative adhesions and ovarian reserve damage [57] and, therefore, more studies are warranted before this technique overtakes the traditional one. However, in case of ULOD, the right ovary should be the selected site because the left ovary seems more prone to the development of post-LOD adhesion if compared with the right one [39]. An interesting alternative that was recently introduced [58] involves the adjustment of energy applied according to the preoperative ovarian volume following the formula described by Sunj et al. [59]. The thermally adjusted ULOD, compared to the fixed-dose BLOD, has been tested only in a single RCT: dose-adjusted ULOD applied to the larger ovary has comparable ovulation and pregnancy rates to fixed-dose BLOD at 3-month follow-up periods, with a decrease in its effectiveness after 6 months [57]. As a consequence, it is still doubtful if the adjusted diathermy is able to ameliorate androgen control and enhance follicular growth, resulting in better reproductive outcomes compared with the fixed dose [60].

The mini-invasive theory has been largely applied to the ovarian drilling procedure. Over the years, laparoscopic surgical procedures have shown refinements in an attempt to reduce abdominal wall trauma, postoperative pain, and hernia formation and to improve cosmesis by decreasing the number of ports or reducing the port size with a shift from larger (10 mm) to a smaller endoscope; additional goals are the possibility of performing ovarian drilling as an outpatient procedure without the need for general anesthesia with comparable results to the more invasive procedure [46].

The feasibility and efficacy of mini-laparoscopy with a 5.0 mm laparoscope and ancillary ports of 3 mm under local anesthesia and conscious sedation have been successfully reported [61]. Beneficial effects in terms of pregnancy rate and faster discharge time (<2 h) render office micro-laparoscopic ovarian drilling (OMLOD) a new modality treatment option to be performed in an outpatient setting, under local anesthesia, with a very low pain score [62]. In conformity with the concept of mini-invasiveness, vaginal access has been proposed as a possible route for ovarian drilling.

Transvaginal hydrolaparoscopy (THL) under general anesthesia using bipolar electrosurgery was first reported by Fernandez et al. as a new approach for ovarian drilling in women with PCOS [63]. Further advances in THL have now allowed ovarian drilling to be performed by the transvaginal route in a day surgery regimen [64]. At present, THL could be considered as an outpatient tool useful to perform in the same time diagnostic infertility investigation and minimal operative procedures with a low complication and failure rate: in a retrospective cohort study of 2288 procedures, of which 374 were ovarian drilling, failure to access the pouch of Douglas occurred in 23 patients (1%). The complication rate was 0.74%, due to bowel perforations (*n* = 13) and bleeding (*n* = 4) requiring laparoscopy. All bowel perforations were treated conservatively, with 6 days of antibiotics, and no further complications occurred [65]. The efficacy of transvaginal hydrolaparoscopic ovarian drilling (THLOD) in terms of ovulation and pregnancy rates in PCOS patients was reported in several studies with comparable results to those of laparoscopy [64,66,67,68]. THLOD has a lower risk of adhesions than LOD (44). This finding was ascribed both to the instillation of saline solution into the peritoneal cavity rather than the irritative action of pneumoperitoneum and the shorter duration of the procedure [69].

Noteworthily, in a case–control study on 123 women with clomiphene-resistant PCOS, the same authors evaluated the effects of THLOD on ovarian volume, power Doppler flow indices and serum AMH levels. A significant reduction was achieved in these parameters as compared to the preoperative values [59]. In addition, THLOD, as already suggested by Ferraretti et al. [34], due to the recovery of ovary sensitivity to gonadotrophins after ovarian drilling, could be considered as an effective low invasive and low-cost method for improvement in ovarian response in PCOS poor responding patients admitted to IVF treatment [70].

In recent years, there have been reports on the role of ultrasound-guided transvaginal ovarian needle drilling (UTND) as a novel surgical method used to induce ovulation for women with clomiphene-resistant PCOS in an outpatient setting. The idea of needle drilling came to mind through the observation of successful spontaneous ovarian ovulatory performance in patients with PCOS after previous follicular aspiration for IVF treatment. A long sharp needle (35 cm—16 gauge) connected to a continuous manual vacuum pressure is used to puncture each ovary from different angles to aspirate all visible small follicles, under the guidance of the ultrasound [71]. Aspiration of follicular fluid through UTND reduces intraovarian and serum androgen and LH levels, rapidly restoring feedback to the hypothalamus and pituitary. It is likely that the removal of other factors, such as inhibin and other intraovarian substances, may also be involved. Preliminary data suggest that UTND is a safe technique and the outcome with regard to the ovulatory and pregnancy rate is comparable to the standard bipolar drilling without the potential risks of thermal damage [72]. With conscious sedation, this procedure is well-tolerated by the patient and can be repeated in case of failed ovulation or recurrent anovulatory states.

## 7. Conclusions

Many therapeutic strategies have been considered for inducing ovulation in the case of CC-resistant patients, but the gold standard for the management of these patients is yet to be identified [73]. LOD is a surgical treatment for anovulatory PCOS women that is as effective as gonadotropins in terms of pregnancy and live birth rates but without the risks of ovarian hyperstimulation syndrome and multiple pregnancies. It offers several advantages over this treatment such as no need for complex monitoring, cost reduction, long-term improved spontaneous resumption of ovulation and menstruation, and more favorable ovarian responsiveness to successive ovulation induction agents. Despite its advantages, owing to the potential risks of iatrogenic adhesions and decreased ovarian reserve, LOD should only be considered in highly selected cases and for the sole purpose of correcting anovulatory infertility. New therapeutic low-degree invasive procedures are slowly replacing the traditional surgical approach. At present, the place of these new therapeutic options is still not clarified and none of them have provided enough evidence in terms of efficacy and safety. Further studies are necessary to encourage these techniques but until these preliminary data will be substantiated and strengthened by high-quality evidence, LOD alone or in association with medical ovulation should be employed after carefully weighing the benefits against the potential risks.

## Figures and Tables

**Table 1 medicina-58-01002-t001:** Ovarian drilling: key factors to success.

Patient should be carefully selected considering that obesity (BMI > 25), low basal luteinizing hormone (LH) (<10 IU/L), duration of infertility > 3 years, marked biochemical hyperandrogenism (free androgen index—FAI > 15) and high basal anti-müllerian hormone AMH (>7.7 ng/mL) are predictors of poor response.
2. The most accredited strategy consists of performing four punctures bilaterally, for a depth of 3–4 m, each for 4 s at 40 W (rule of 4) delivering 640 J of energy per ovary.
3. Mini laparoscopy with a 5.0 mm laparoscope and ancillary ports of 3 mm under regional anesthesia could be employed to ensure a faster recovery and better cosmetic results.
4. Before the application of energy, the ovary should be carefully lifted away from the intestine and ureters.
5. Peritoneal cavity and ovaries should be cooled using up to 1000 mL of isotonic solution to heat lesions and reduce the risk of post-operative adhesion formation.

**Table 2 medicina-58-01002-t002:** Ovarian drilling indications in anovulatory infertile women with polycystic ovary syndrome (PCOS).

First-line
When additional reasons justify laparoscopic surgery (e.g., diagnostic evaluation for tubal patency, uterine malformation)
Second-line
Subcutaneous gonadotropin therapy, despite the risk of long-lasting effects, multiple pregnancies and hyperstimulation syndrome
Third-line
In case of failure of medical therapy To enhance ovarian response to medical therapy

## Data Availability

The data presented in this study are available on PubMed, Medline and Embase.

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
