# Peer review of "Ovarian Drilling: Back to the Future"

_medicina, 2022, doi:10.3390/medicina58081002_

Round 1

Reviewer 1 Report

I find the topic that is reviewed is stimulating and even provoking, and  is well written and presented.

They presented profoundly the traditional and modern technic for ovarian drilling.

Table 1 : paragraphs 6-10 are like 1-5 is it so ?!

The main recommendations thar should be considered for any women  (table 2)  is that operative manipulation of the ovaries  should be suggested  as 1st line therapy in case of any laparoscopic surgery, moreover as the second line therapy after failure of oral agents instead of gonadotropins, and according to this table as one step even before proceeding to ART.

According to ref 13: Gonadotrophins could be used as second line pharmacological agents in women with PCOS who have failed first line oral ovulation induction therapy and are anovulatory and infertile, with no other infertility factors .

Author Response

Thank you for your global positive comments.

Point 1: Table 1: paragraphs 6-10 are like 1-5 is it so ?!

Response 1: Table 1 has been corrected. Paragraphs repeated have been removed.

Point 2: The main recommendations that should be considered for any women (table 2)  is that operative manipulation of the ovaries  should be suggested  as 1st line therapy in case of any laparoscopic surgery, moreover as the second line therapy after failure of oral agents instead of gonadotropins, and according to this table as one step even before proceeding to ART. According to ref 13: Gonadotrophins could be used as second line pharmacological agents in women with PCOS who have failed first line oral ovulation induction therapy and are anovulatory and infertile, with no other infertility factors 

Response 2: We thank the reviewer for pointing this out. Ovarian drilling has to be considered 1st line therapy in case surgery should already be performed for any other additional reasons. This should apply only to women with anovulatory infertility for whom a diagnosis of PCOS has been made. To avoid misunderstanding we have specified this in the title of table 2.  

Reviewer 2 Report

The article "Ovaraian Drilling: back to the future" presents an interesting review of the potential use of laparoscopic ovarian drilling and its modification in ovulation induction in polycystic ovary syndrome patients.

The review elegantly presents possible strategies for infertility treatment in PCOS.

The are unnecessary repetitions in the Table 1 - that need to be corrected. Moreover, some references are listed twice (reference number 2 and 7)

The authors in line 71-72 give the information form the reference number 7 that the eficacy of ovulation induction with clomifene citrate is around 10-50%. According to my knowledge the eficacy of ovulation induction with clomifene citrate is higher than that. Moreover, I could not find such infromation in the quoted citation. This needs explanation.

My overall recommendation is to reconsider the article after major revision

Author Response

First of all, thank you for your feedback: it will increase surely the quality and readability of the paper.

Point 1: The are unnecessary repetitions in the Table 1 - that need to be corrected.

Response 1: Repetitions have been corrected

Point 2: Some references are listed twice (reference number 2 and 7)

Response 2: References repetition has been corrected

Point 3: The authors in line 71-72 give the information form the reference number 7 that the eficacy of ovulation induction with clomifene citrate is around 10-50%. According to my knowledge the efficacy of ovulation induction with clomifene citrate is higher than that. Moreover, I could not find such information in the quoted citation. This needs explanation.

Response 3: Thank you very much for the reminder.

The efficacy of ovulation induction with clomifene citrate is definitively higher than 10-50%. We wanted to emphasize its efficacy stating that almost 50 % of patients will respond with a dosage of 50mg. Of the remaining patients, an additional 20% will respond with a  dosage increase to 100mg. Among patients who did not respond, another 10% will success with a dosage of 150 mg. This is in accordance with the “step-up” protocol of clomiphene citrate. In order to avoid misinterpretations, however, the sentence has been reformulated and the reference has been replaced with another that clearly states this concept (Lines 73-76)

Round 2

Reviewer 1 Report

The author didn't respond to my comment concerning the table which state that Ovarian drilling is indicted as second line therapy after failure of oral agents.

  I disagree with this statement, the second line therapy is s.c gonadotropins(also according to ref 13 in the manuscript)

Author Response

I'am sorry if we are not answered to your previous request.

Now, we have edited the table 2 reporting the proper second-line indication.